# Domain Generalization via Entropy Regularization

**Shanshan Zhao**
The University of Sydney
Australia
szha4333@uni.sydney.edu.au

**Mingming Gong**
University of Melbourne
Australia
mingming.gong@unimelb.edu.au

**Tongliang Liu**
The University of Sydney
Australia
tongliang.liu@sydney.edu.au

**Huan Fu**
Alibaba Group
China
fuhuan.fh@alibaba-inc.com

**Dacheng Tao**
The University of Sydney
Australia
dacheng.tao@sydney.edu.au

## Abstract

Domain generalization aims to learn from multiple source domains a predictive model that can generalize to unseen target domains. One essential problem in domain generalization is to learn discriminative domain-invariant features. To arrive at this, some methods introduce a domain discriminator through adversarial learning to match the feature distributions in multiple source domains. However, adversarial training can only guarantee that the learned features have invariant marginal distributions, while the invariance of conditional distributions is more important for prediction in new domains. To ensure the conditional invariance of learned features, we propose an *entropy regularization* term that measures the dependency between the learned features and the class labels. Combined with the typical task-related loss, *e.g.,* cross-entropy loss for classification, and adversarial loss for domain discrimination, our overall objective is guaranteed to learn conditional-invariant features across all source domains and thus can learn classifiers with better generalization capabilities. We demonstrate the effectiveness of our method through comparison with state-of-the-art methods on both simulated and real-world datasets. Code is available at: https://github.com/sshan-zhao/DG_via_ER.

## 1 Introduction

Recent years have witnessed the remarkable success of modern machine learning techniques in various applications. However, a fundamental problem machine learning suffers from is that the model learned from training data often does not generalize well on data sampled from a different distribution, due to the existence of data bias [1, 2] between the training and test data. To tackle this issue, a significant effort has been made in domain adaptation, which reduces the discrepancy between source and target domains [3–8]. The main drawback of this approach is that one has to repeat training for each new dataset, which can be time-consuming. Therefore, *domain generalization* [9] is proposed to learn generalizable models by leveraging information from multiple source domains [10–13].

Since there is no prior information about the distribution of the target domain during training, it is difficult to match the distributions between source and target domains, which makes domain

generalization more challenging. To improve the generalization capabilities of learned models, various solutions have been developed from different perspectives. A classic but effective solution to domain generalization is learning a domain-invariant feature representation [11, 12, 14, 10, 15, 14] across source domains. Muandet *et al.* [10] presented a kernel-based optimization algorithm, called Domain-Invariant Component Analysis, to learn an invariant transformation by minimizing the dissimilarity across domains. Ghifary *et al.* [11] proposed to learn features robust to variations across domains by introducing multi-task auto-encoders. Another line of research explores various data augmentation strategies [16–18]. For example, Shankar *et al.* [16] presented a gradient-based domain perturbation strategy to perturb the input data. By augmenting the original feature space, Blanchard *et al.* [19] viewed the problem of domain generalization as a kind of supervised learning problem. Then, they developed a kernel-based method that predicts classifiers from the augmented feature space. To make theoretical complementary to these empirically supported approaches, Deshmukh *et al.* [20] proved the first known generalization error bound for multi-class domain generalization through studying a kernel-based learning algorithm. Apart from the clues aforementioned, some recent works [21–24] attempted to exploit meta-learning for domain generalization. A latest work, MASF [21], proposed a model-agnostic episodic learning procedure to regularize the semantic structure of the feature space.

In this paper, we revisit the domain-invariant feature representation learning methods. Most of existing methods assume that the marginal distribution $P(X)$ changes while the conditional distribution $P(Y|X)$ stays stable across domains. Therefore, significant effort has been made in learning a feature representation $F(X)$ that has invariant $P(F(X))$, either by traditional moment matching [25] or modern adversarial training [15, 14]. To ensure the universality of $F(X)$ and also make it discriminative, a joint classification model is trained on all the source domains and can be used for prediction in new datasets. However, the stability of $P(Y|X)$ is often violated in real applications, leading to sub-optimal solutions. Li *et al.* [14] proposed to learn invariant class-conditional distribution $(P(F(X)|Y))$ by doing adversarial training for each class. However, the method becomes less effective as the number of classes increases.

To tackle the aforementioned issues, we propose an entropy-regularization approach which directly learns features that have invariant $P(Y|F(X))$ across domains. In specific, the conditional entropy term $H(Y|F(X))$ measures the dependency between $F(X)$ and class label $Y$, and we aim to minimize the dependency by maximizing the conditional entropy. We show theoretically that our entropy-regularization together with the cross-entropy classification loss effectively minimize the divergence between $P(Y|F(X))$ in all source domains. In addition, we show that $H(Y|F(X))$ can be effectively estimated by assuming a multinomial distribution for $P(Y|F(X))$, which is a weak assumption for discrete class labels. Together with the adversarial training on $P(F(X))$, our approach can guarantee the invariance of the joint distribution $P(F(X), Y)$ and thus has a better generalization capability. We demonstrate the effectiveness of our approach through conducting comprehensive experiments on several benchmark datasets.

## 2 Method

### 2.1 Problem Definition

Let $\mathcal{X}$ and $\mathcal{Y}$ be the feature and label spaces, respectively. In the domain generalization subject, there are $K$ source domains $\{\mathcal{D}_i\}_{i=1}^{K}$ and $L$ target domains[1] $\{\mathcal{D}_i\}_{i=K+1}^{L+K}$. The goal is to generalize the model learned using data samples of source domains to unseen target domains. In the following, we denote the joint distribution of domain $i$ by $P_i(X, Y)$ (defined on $\mathcal{X} \times \mathcal{Y}$). During the training process, there are $K$ datasets $\{S_i\}_{i=1}^{K}$ available, where $S_i = \{(\mathbf{x}_j^{(i)}, y_j^{(i)})\}_{j=1}^{N_i}$. Here, $N_i$ is the number of samples of $S_i$, which are sampled from the $i^{th}$ domain. In the test stage, we evaluate the generalization capabilities of the learned model on $L$ datasets sampled from the $L$ target domains, respectively. This paper mainly studies domain generalization for image classification, where the label space $\mathcal{Y}$ contains $C$ discrete labels $\{1, 2, \cdots, C\}$.

## 2.2 Domain Generalization Through Adversarial Learning

We first present how domain generalization can be learned in an adversarial learning framework.

For the classification subject, the model consists of one feature extractor $F$ parameterized by $\theta$ and one classifier $T$ parameterized by $\phi$. We can optimize $\theta$ and $\phi$ on the $K$ source datasets by minimizing a cross-entropy loss:

$$
\begin{aligned}
\min_{F,T} \mathcal{L}_{cls}(\theta, \phi) &= -\sum_{i=1}^{K} \mathbb{E}_{(X,Y) \sim P_i(X,Y)} [\log(Q^T(Y|F(X)))] \\
&= -\sum_{i=1}^{K} \sum_{j=1}^{N_i} \mathbf{y}_j^{(i)} \cdot \log(T(F(\mathbf{x}_j^{(i)}))),
\end{aligned}
\tag{1}
$$

where $\mathbf{y}_j^{(i)}$ is the one-hot vector of the class label $y_j^{(i)}$, "$\cdot$" represents the dot product operation, and $Q^T(Y|F(X))$ denotes the predicted label distribution (conditioned on $F(X)$) corresponding to domain $i$.

However, optimized by the classification loss solely, the model cannot learn domain-invariant features, and thus shows limitations in generalizing to the unseen domains. By exploiting the adversarial learning [26], we can alleviate the issue. Specifically, we further introduce a domain discriminator $D$ parameterized by $\psi$, and train $D$ and $F$ in a minimax game as follows:

$$
\begin{aligned}
\min_{F} \max_{D} \mathcal{L}_{adv}(\theta, \psi) &= \sum_{i=1}^{K} \mathbb{E}_{X \sim P_i(X)} [\log D(F(X))] \\
&= \sum_{i=1}^{K} \sum_{j=1}^{N_i} \mathbf{d}_j^{(i)} \cdot \log(D(F(\mathbf{x}_j^{(i)}))),
\end{aligned}
\tag{2}
$$

where $\mathbf{d}_j^{(i)}$ is the one-hot representation of the domain label $i$.

Although optimizing Eq. 2 can lead to invariant marginal distributions *i.e.,* $P_1(F(X)) = P_2(F(X)) = \cdots = P_K(F(X))$, it cannot guarantee the conditional distribution $P(Y|F(X))$ is invariant across domains. This would degrade the generalization capabilities of the model. Even though the classifier attempts to cluster the samples from the same category together in the feature space, which benefits to the learning of the invariant conditional distribution, there still exists an issue. We take the simulated data for example. Firstly, we sample data from two 2D-distributions (shown in Figure 1) as the Domain_0 and Domain_1, respectively. The marginal distributions of the first dimension ($x_0$) in the two domain are the same, while the second ($x_1$) comes from different marginal distributions. Each domain consists of three components. We take each dimension as the input to train a classifier using Eq. 1 and Eq. 2, and we find that the classifier distinguishes the second dimension better than the first (loss: $-0.34$ *v.s.* $-0.16$). This indicates that the classifier might not select the domain-invariant feature, but select the features easier to discriminate. Therefore, it is challenging for the typical classification loss to achieve a balance between learning domain-invariant features and discriminative features.

## 2.3 Entropy Regularization

**Description.** To address the issues aforementioned, we propose to regularize the distributions of the features by minimizing the KL divergence between the conditional distribution $P_i(Y|F(X))$ in the $i^{th}$ domain and the conditional distribution $Q^T(Y|X)$. $P_i(Y|F(X))$ denotes the predicted label distribution conditioned on the learned features. By matching any conditional distribution $P_i(Y|F(X))$ to a common distribution $Q^T(Y|F(X))$, we can obtain the domain-invariant conditional distribution $P(Y|F(X))$. For the purpose, we define an optimization problem as follows:

$$
\min_{F,T} \sum_{i=1}^{K} KL(P_i(Y|F(X)) || Q^T(Y|F(X))).
\tag{3}
$$

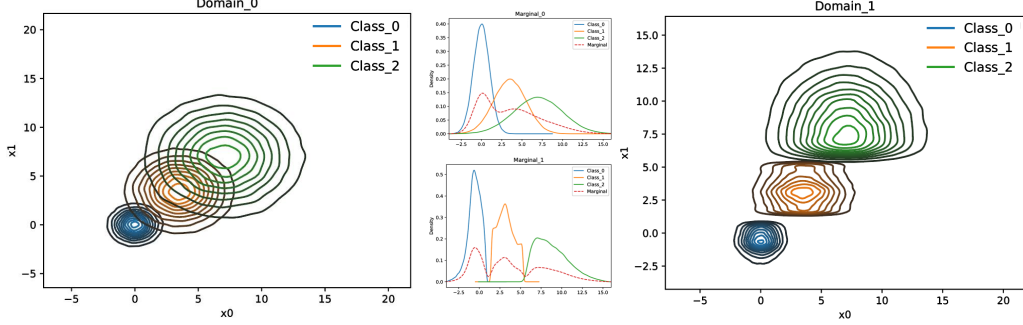

Figure 1: Simulated data. We create two domains from the two 2D-distributions (left and right), respectively. The data in Domain_0 and Domain_1 is two-dimensional. In specific, the first dimensions in two domains are both sampled from Marginal_0 (top-middle), while the second dimension in Domain_0 and Domain_1 is sampled from Marginal_0 and Marginal_1 (bottom-middle), respectively.

By using the definition of the KL divergence, we have:

$$
\min_{F,T} \sum_{i=1}^{K} KL(P_i(Y|F(X))||Q^T(Y|F(X))) = \sum_{i=1}^{K} \mathbb{E}_{(X,Y)\sim P_i(X,Y)} [\log \frac{P_i(Y|F(X))}{Q^T(Y|F(X))}]
$$

$$
= \sum_{i=1}^{K} \mathbb{E}_{(X,Y)\sim P_i(X,Y)} [\log P_i(Y|F(X))] - \sum_{i=1}^{K} \mathbb{E}_{(X,Y)\sim P_i(X,Y)} [\log Q^T(Y|F(X))].
$$
(4)

The second term is actually the cross-entropy classification loss (Eq. 1), while the first one is the sum of $K$ negative conditional entropy terms $\sum_{i=1}^{K} -H_{P_i}(Y|F(X))$. However, it is difficult to optimize $-H_{P_i}(Y|F(X))$ directly, since we do not know the conditional distribution $P_i(Y|F(X))$. To overcome this issue, we first provide the following theorem to exploit the relationship between the negative conditional entropy term and the Jensen-Shannon divergence (JSD) between the conditional distributions $\{P_i(F(X)|Y=c)\}_{c=1}^{C}$.

**Theorem 1.** *Assuming that all classes are equally likely, minimizing $-H_{P_i}(Y|F(X))$ is equivalent to minimizing the JSD between the conditional distributions $\{P_i(F(X)|Y=c)\}_{c=1}^{C}$. The global minimum is achieved if and only if $P_i(F(X)|Y=1) = P_i(F(X)|Y=2) = \cdots = P_i(F(X)|Y=C)$. Note that, if the dataset is balanced, it is easy to make the assumption satisfied. Otherwise, we can enforce it through biased batch sampling.*

The proof is given in Sec. S1 of the Supplementary Materials. Inspired by Theorem 1 and the minimax game proposed in GAN [26] and conditional GAN [27], we introduce $K$ additional classifiers $\{T_i'\}_{i=1}^{K}$, and then present the following minimax game:

$$
\min_{F} \max_{\{T_i'\}_{i=1}^{K}} V(F,T_1',T_2',\cdots,T_K') = \sum_{i=1}^{K} \mathbb{E}_{(X,Y)\sim P_i(X,Y)} [\log Q_i^{T_i'}(Y|F(X))],
$$
(5)

where $T_i'$ parameterized by $\phi_i'$ represents a classifier trained on data sampled from domain $\mathcal{D}_i$, and $Q_i^{T_i'}(Y|F(X))$ denotes the conditional distribution induced by $T_i'$. The following theorem (the proof can be found in Sec. S2 of the Supplementary Materials) shows that the minimax game is equal to minimizing the JSD between the conditional distributions $\{P_i(F(X)|Y=c)\}_{c=1}^{C}$. According to Theorem 1, we can thus achieve the optimization of $\sum_{i=1}^{K} -H_{P_i}(Y|F(X))$.

**Theorem 2.** *If $U(F)$ is the maximum value of $V(F,T_1',T_2',\cdots,T_K')$, i.e.,*

$$
U(F) = \max_{\{T_i'\}_{i=1}^{K}} V(F,T_1',T_2',\cdots,T_K'),
$$
(6)

*the global minimum of the minimax game is attained if and only if $P_i(F(X)|Y=1) = P_i(F(X)|Y=2) = \cdots = P_i(F(X)|Y=C)$. At this point, $U(F)$ attains the value $-KC\log C$.*

Therefore, our proposed entropy regularization loss can be defined as:

$$
\min_{F} \max_{\{T_i'\}_{i=1}^{K}} \mathcal{L}_{er}(\theta, \{\phi_i'\}_{i=1}^{K}) = \sum_{i=1}^{K} \mathbb{E}_{(X,Y)\sim P_i(X,Y)} [\log Q_i^{T_i'}(Y|F(X))].
$$
(7)

Combining Eq. 7 with the classification loss (Eq. 1) and the domain discrimination loss (Eq. 2), we obtain the training objective:

$$\min_{F,T} \max_{D,\{T_i'\}_{i=1}^K} \mathcal{L}(\theta, \phi, \psi, \{\phi_i'\}_{i=1}^K) = \mathcal{L}_{cls}(\theta, \phi) + \alpha_1 \mathcal{L}_{adv}(\theta, \psi) + \alpha_2 \mathcal{L}_{er}(\theta, \{\phi_i'\}_{i=1}^K), \quad (8)$$

where $\alpha_1$ and $\alpha_2$ are trade-off parameters.

**Algorithm.** In our experiments, we observed that directly optimizing the loss Eq. 8 may show instability, since the minimax game in Eq. 7 encourages the learned features not to be distinguished by the classifiers. That may impede the optimization of the classification loss. To alleviate this issue, we introduce additional classifiers $\{T_i\}_{i=1}^K$ and add a new cross-entropy loss $\mathcal{L}_{cel}$:

$$\min_{F,\{T_i\}_{i=1}^K} \mathcal{L}_{cel}(\theta, \{\phi_i\}_{i=1}^K) = - \sum_{i=1}^K \mathbb{E}_{(X,Y) \sim P_i(X,Y)} [\log Q_i^{T_i}(Y|\bar{F}(X))]$$
$$- \sum_{i=1}^K \sum_{j=1, j \neq i}^K \mathbb{E}_{(X,Y) \sim P_j(X,Y)} [\log Q_i^{\bar{T}_i}(Y|F(X))], \quad (9)$$

where $Q_i^{T_i}(Y|F(X))$ denotes the conditional distribution induced by $T_i$. Here, $\bar{F}$ and $\bar{T}_i$ mean that we fix the parameters of $F$ and $T$ during the training procedure, respectively. Specifically, we feed the learned features in the $i^{th}$ domain into $T_i$ to optimize its parameters $\phi_i$. Additionally, we expect the feature extractor can map the data in domains $\{\mathcal{D}_j\}_{j=1,j \neq i}^K$ to a representation, which can be distinguished by $T_i$ accurately. This strategy, on the one hand, can impose regularization on the feature distribution of domains $\{\mathcal{D}_j\}_{j=1,j \neq i}^K$. On the other hand, the new loss can be considered as a complementary of $\mathcal{L}_{cls}$.

Thus, our final objective is formulated as:

$$\min_{F,T,\{T_i\}_{i=1}^K} \max_{D,\{T_i'\}_{i=1}^K} \mathcal{L}(\theta, \phi, \psi, \{\phi_i\}_{i=1}^K, \{\phi_i'\}_{i=1}^K) = \mathcal{L}_{cls} + \alpha_1 \mathcal{L}_{adv} + \alpha_2 \mathcal{L}_{er} + \alpha_3 \mathcal{L}_{cel}, \quad (10)$$

where $\alpha_3$ is a weighting factor. To illustrate the training process clearly, we provide the pseudo-code of our algorithm in Alg. 1. We also provide the framework in the Supplementary Materials.

---

**Algorithm 1:** Training algorithm for domain generalization via entropy regularization.

---

**Input:** $\{S_i\}_{i=1}^K$: $K$ source training datasets
**Input:** $\alpha_1, \alpha_2, \alpha_3$: weighting factors
**Output:** $F$: feature extractor; $T, \{T_i\}_{i=1}^K, \{T_i'\}_{i=1}^K$: classifier; $D$: discriminator
**while** *training is not end* **do**
    Sample data from each training dataset respectively
    Update $\theta$, $\phi$, and $\psi$ by optimizing the first and second terms of Eq. 10
    **for** $i$ *in* $1 : K$ **do**
        Sample data from the $i^{th}$ dataset $S_i$
        Update $\{\phi_i\}_{i=1}^K$ by optimizing the forth term of Eq. 10
        Update $\theta$, and $\{\phi_i'\}_{i=1}^K$ by optimizing the third term of Eq. 10
        Sample data from datasets $\{S_j\}_{j=1,j \neq i}^K$
        Update $\theta$ by optimizing the forth term of Eq. 10.
    **end**
**end**

---

**Discussion.** In comparison with the typical classification loss, our entropy regularization loss can push the network to learn domain-invariant features. For instance, in the example of simulated data in Figure 1, the summation of the classification loss, the regularization loss and the domain adversarial loss is $-0.16$ in classifying the first dimension, and is $-0.02$ in classifying the second dimension. Therefore, our training objective can enforce the learned features to be domain-invariant.

## 3 Experiments

In this section, we study domain generalization on four datasets, including two simulated datasets (*i.e.,* Rotated MNIST [11] and Rotated CIFAR-10) and two real-world datasets (*i.e.,* VLCS [11],

Table 1: Results on MNIST dataset with object recognition accuracy (%) averaged over 10 runs.

| Target | CrossGrad [16] | MetaReg [23] | Reptile [34] | Feature-Critic [30] | DeepAll | Basic-Adv | Ours |
|--------|---------------|--------------|--------------|---------------------|---------|-----------|------|
| $M_0$ | 86.03 | 85.70 | 87.78 | 87.04 | $88.37 \pm 1.19$ | $88.88 \pm 1.08$ | $\mathbf{90.09 \pm 1.25}$ |
| $M_{15}$ | 98.92 | 98.87 | 99.44 | $\mathbf{99.53}$ | $99.13 \pm 0.41$ | $99.10 \pm 0.19$ | $99.24 \pm 0.37$ |
| $M_{30}$ | 98.60 | 98.32 | 98.42 | $\mathbf{99.41}$ | $99.28 \pm 0.27$ | $99.25 \pm 0.14$ | $99.27 \pm 0.16$ |
| $M_{45}$ | 98.39 | 98.58 | 98.80 | $\mathbf{99.52}$ | $99.09 \pm 0.29$ | $99.25 \pm 0.17$ | $99.31 \pm 0.21$ |
| $M_{60}$ | 98.68 | 98.93 | 99.03 | 99.23 | $99.14 \pm 0.28$ | $99.16 \pm 0.32$ | $\mathbf{99.45 \pm 0.19}$ |
| $M_{75}$ | 88.94 | 89.44 | 87.42 | $\mathbf{91.52}$ | $87.48 \pm 1.01$ | $89.06 \pm 1.54$ | $90.81 \pm 1.35$ |
| $Avg.$ | 94.93 | 94.97 | 95.15 | 96.04 | 95.42 | 95.78 | $\mathbf{96.36}$ |

Table 2: Results on CIFAR-10 dataset with object recognition accuracy (%) averaged over 5 runs.

| Method | $M0$ | $M15$ | $M30$ | $M45$ | $M60$ | $M75$ | $Avg.$ |
|--------|------|-------|-------|-------|-------|-------|--------|
| DeepAll | $71.28 \pm 1.59$ | $97.94 \pm 0.32$ | $99.14 \pm 0.04$ | $99.06 \pm 0.19$ | $99.07 \pm 0.40$ | $76.59 \pm 0.89$ | 90.51 |
| Basic-Adv | $75.85 \pm 1.45$ | $99.03 \pm 0.18$ | $99.16 \pm 0.06$ | $99.14 \pm 0.11$ | $99.29 \pm 0.13$ | $\mathbf{81.14 \pm 1.34}$ | 92.27 |
| Ours | $\mathbf{77.91 \pm 0.83}$ | $\mathbf{99.05 \pm 0.22}$ | $\mathbf{99.33 \pm 0.09}$ | $\mathbf{99.39 \pm 0.14}$ | $\mathbf{99.40 \pm 0.29}$ | $80.12 \pm 0.60$ | $\mathbf{92.53}$ |

PACS [28]). We make comparisons against state-of-the-art methods to demonstrate the effectiveness of the proposed algorithm. We conduct extensive ablations to discuss our method comprehensively.

## 3.1 Simulated Datasets

**Rotated MNIST.** Following the setting in [11], we first randomly choose 100 samples per category (1000 in total) from the original dataset [29] to form the domain $M_0$. Then, we create 5 rotating domains $\{M_{15}, M_{30}, M_{45}, M_{60}, M_{75}\}$ by rotating each image in $M_0$ five times with 15 degrees intervals in clock-wise direction. As done by previous works [30, 16], we conduct leave-one-domain-out experiments by selecting one domain to hold out as the target. For fair comparisons, we exploit the standard MNIST CNN, where the feature network consists of two convolutional layers and one fully-connected (FC) layer, and the classifier has one FC layer. We train our model with the learning rate of $1e-4$ ($F$, $T$, and $D$), and $1e-5$ ($\{T_i, T_i'\}_{i=1}^5$) for $3,000$ iterations. We set the weighting factors to $0.5$ ($\alpha_1$), $0.005$ ($\alpha_2$), and $0.01$ ($\alpha_3$), respectively. We repeat all of the experiments 10 times, and report the average mean and standard deviation of recognition accuracy in Table 1.

**Rotated CIFAR-10.** We randomly choose 500 samples per category (5000 in total) from the original CIFAR-10 dataset [31], and then create additional 5 domains using the same strategy as stated in Rotated MNIST. We use AlexNet [32] as our backbone network. In specific, the feature extractor $F$ consists of the top layers of AlexNet model till the POOL5 layer, while $T$ contains FC6, FC7, and an additional FC layer. For $\{T_i, T_i'\}_{i=1}^5$ and $D$, we use a similar architecture to $T$. We train the whole network *from scratch* with the learning rate of $1e-3$ ($F$, $T$, and $D$) and $1e-4$ ($\{T_i, T_i'\}_{i=1}^5$) using the Adam optimizer [33] for 2000 iterations. The weighting factors ($\alpha_1$, $\alpha_2$, $\alpha_3$) are set to 0.5, 0.001, and 0.1, respectively. We repeat all experiments 5 times, and provide the results in Table 2.

**Results.** We make comparisons against several recent works, *e.g.,* CrossGrad [16], MetaReg [23], Reptile [34], and Feature-Critic [30], on Rotated MNIST. To better illustrate the generalization capabilities of our model, we also evaluate the performance of two additional models, *i.e.,* DeepAll and Basic-Adv, on both Rotated MNIST and Rotated CIFAR-10. DeepAll trains $F$ and $T$ on all of the source domains without performing any domain generalization (Eq. 1), while Basic-Adv is the basic solution through adversarial learning (Eq. 1 and Eq. 2). We can find all of the algorithms perform well on Rotated MNIST from Table 1, which means the generated domains have similar distributions. Nevertheless, our approach still performs better than existing approaches. Furthermore, the higher accuracy compared with DeepAll and Basic-Adv on both Rotated MNIST and Rotated CIFAR-10 shows the better generalization capabilities of the proposed algorithm.

## 3.2 Real-World Datasets

**VLCS.** VLCS [11] contains images from four well-known datasets, *i.e.,* Pascal VOC2007 (V) [37], LabelMe (L) [38], Caltech (C) [39], and SUN09 (S) [40]. There are five categories, including bird, car, chair, dog, and person. Following previous works [11, 22, 21], we randomly split each domain data into training (70%) and test (30%) sets, and do the leave-one-out evaluation. For the configuration of the network, we consider two cases, *i.e.,* MLP and E2E. In specific, in MLP, we use the pre-extracted DeCAF6 features (4096-dimensional vector) as the input, and $F$ consists of two FC layers with latent

Table 3: Results on VLCS dataset with object recognition accuracy (%) averaged over 20 runs.

| Method | Pascal VOC2007 | LabelMe | Caltech | SUN09 | Average |
|---|---|---|---|---|---|
| MLP | | | | | |
| D-MATE [11] | 63.90 | 60.13 | 89.05 | 61.33 | 68.60 |
| DBADG [28] | 65.58 | 58.74 | 92.43 | 61.85 | 69.65 |
| CCSA [35] | 67.10 | 62.10 | 92.30 | 59.10 | 70.15 |
| MetaReg [23] | 65.00 | 60.20 | 92.30 | 64.20 | 70.43 |
| CrossGrad [16] | 65.50 | 60.00 | 92.00 | 64.70 | 70.55 |
| DANN [36] | 66.40 | 64.00 | 92.60 | 63.60 | 71.65 |
| MMD-AAE [12] | 67.70 | 62.60 | 94.40 | 64.40 | 72.28 |
| MLDG [24] | 67.70 | 61.30 | 94.40 | 65.90 | 72.33 |
| Epi-FCR [22] | 67.10 | **64.30** | 94.10 | 65.90 | 72.85 |
| DeepAll | $70.07 \pm 0.79$ | $60.54 \pm 1.02$ | $93.83 \pm 1.08$ | $65.95 \pm 1.13$ | 72.60 |
| Basic-Adv | $70.47 \pm 0.59$ | $60.94 \pm 0.94$ | $93.84 \pm 1.00$ | $66.05 \pm 0.91$ | 72.82 |
| Ours | $\mathbf{70.54 \pm 0.55}$ | $60.81 \pm 1.38$ | $\mathbf{94.44 \pm 0.98}$ | $\mathbf{66.11 \pm 0.75}$ | **72.97** |
| E2E | | | | | |
| DBADG [28] | 69.99 | 63.49 | 93.64 | 61.32 | 72.11 |
| JiGen [18] | 70.62 | 60.90 | 96.93 | 64.30 | 73.19 |
| MMLD [15] | 71.96 | 58.77 | 96.66 | 68.13 | 73.88 |
| CIDDG [14] | 73.00 | 58.30 | 97.02 | 68.89 | 74.30 |
| DeepAll | $73.11 \pm 0.67$ | $58.07 \pm 0.52$ | $\mathbf{97.15 \pm 0.40}$ | $68.79 \pm 0.44$ | 74.28 |
| Basic-Adv | $72.79 \pm 0.67$ | $\mathbf{58.53 \pm 0.69}$ | $97.00 \pm 0.50$ | $68.70 \pm 0.69$ | 74.26 |
| Ours | $\mathbf{73.24 \pm 0.49}$ | $58.26 \pm 0.82$ | $96.92 \pm 0.40$ | $\mathbf{69.10 \pm 0.46}$ | **74.38** |

Table 4: Results on PACS dataset with object recognition accuracy (%) averaged over 5 runs.

| Method | Art Painting | Cartoon | Photo | Sketch | Average |
|---|---|---|---|---|---|
| D-MATE [11] | 60.27 | 58.65 | 91.12 | 47.68 | 64.48 |
| CrossGrad [16] | 61.00 | 67.20 | 87.60 | 55.90 | 67.93 |
| DBADG [28] | 62.86 | 66.97 | 89.50 | 57.51 | 69.21 |
| MLDG [24] | 66.23 | 66.88 | 88.00 | 58.96 | 70.01 |
| Epi-FCR [22] | 64.70 | 72.30 | 86.10 | 65.00 | 72.03 |
| Feature-Critic [30] | 64.89 | 71.72 | 89.94 | 61.85 | 71.20 |
| CIDDG [14] | 66.99 | 68.62 | 90.19 | 62.88 | 72.20 |
| MetaReg [23] | 69.82 | 70.35 | 91.07 | 59.26 | 72.62 |
| JiGen [18] | 67.63 | 71.71 | 89.00 | 65.18 | 73.38 |
| MMLD [15] | 69.27 | **72.83** | 88.98 | 66.44 | 74.38 |
| MASF [21] | 70.35 | 72.46 | **90.68** | 67.33 | 75.21 |
| DeepAll | $68.35 \pm 0.80$ | $70.14 \pm 0.87$ | $90.83 \pm 0.32$ | $64.98 \pm 1.92$ | 73.57 |
| Basic-Adv | $71.34 \pm 0.81$ | $70.11 \pm 1.18$ | $88.86 \pm 0.50$ | $70.91 \pm 0.94$ | 75.31 |
| Ours | $\mathbf{71.34 \pm 0.87}$ | $70.29 \pm 0.77$ | $89.92 \pm 0.42$ | $\mathbf{71.15 \pm 1.01}$ | **75.67** |

dimensions of 1024 and 128. For the classifiers $T$ and $\{T_i, T'_i\}_{i=1}^3$, we use one FC layer, respectively. For the discriminator $D$, we utilize three FC layers with the output dimensions of 128, 64, and 3 (the number of source domains). In this case, we train our model with the learning rate of $1e-3$ for 30 epochs using the SGD optimizer. We set all trade-off parameters to $0.1$. In another setting (E2E), we employ the same network configuration as used on Rotated CIFAR-10, but use the model pre-trained on ImageNet [32]. We set the learning rate to $1e-4$, and the weighting factors $\alpha_1$, $\alpha_2$, and $\alpha_3$ to $0.1$, $0.001$, and $0.05$, respectively. We train the model with the batch size of $64$ for each source domain for 60 epochs and repeat all of the experiments 20 times.

**PACS.** PACS [28] is proposed specially for domain generalization. It contains four domains, *i.e.,* Photo (P), Art Painting (A), Cartoon (C), and Sketch (S), and seven categories: dog, elephant, giraffe, guitar, house, horse, and person. For a fair comparison, we use the same training and validation split as presented in [28]. Our network configuration is the same as that used for VLCS (E2E), and we set the weighting factors to $0.5$ ($\alpha_1$), $0.01$ ($\alpha_2$), and $0.05$ ($\alpha_3$), respectively. Then we train the model with the learning rate of $1e-3$ ($F$, $T$, $D$) and $1e-4$ ($\{T_i, T'_i\}_{i=1}^3$) for 60 epochs. We repeat all experiments 5 times, and report the results in Tabel 4.

**Results.** As shown in Table 3, although the baselines (DeepAll and Basic-Adv) are competitive with previous methods in both cases (MLP and E2E), our proposed entropy regularization still improves the performance further on VLCS. Furthermore, the highest average score and the highest score on several domains of PACS can also demonstrate the effectiveness of our approach. For example, Table 4 shows that our method improves the average accuracy by $2.1\%$ on PACS over DeepAll, and improves $6.17\%$ and $2.99\%$ on Sketch and Art Painting, respectively. In addition, from the results in Table 3 and Table 4, we can observe that the performance (Ours *v.s.* DeepAll and Basic-Adv *v.s.* DeepAll) gains obtained by our regularization policy on PACS are more notable than those on VLCS. A possible reason we guess is that only one domain (C) in VLCS is object-centric, while others are

Table 5: Results with different weighting factors on PACS.

| $\alpha_1, \alpha_2, \alpha_3$ | Art Painting | Cartoon | Photo | Sketch | Average |
|---|---|---|---|---|---|
| - , - , - | $68.35 \pm 0.80$ | $70.14 \pm 0.87$ | $90.83 \pm 0.32$ | $64.98 \pm 1.92$ | **73.57** |
| 1.0 , - , - | $64.46 \pm 3.80$ | $64.07 \pm 3.01$ | $83.48 \pm 1.39$ | $66.70 \pm 2.64$ | 69.68 |
| 0.5 , - , - | $71.35 \pm 0.81$ | $70.11 \pm 1.18$ | $88.86 \pm 0.50$ | $70.91 \pm 0.94$ | **75.31** |
| 0.1 , - , - | $68.22 \pm 0.89$ | $70.13 \pm 0.67$ | $90.60 \pm 0.37$ | $64.61 \pm 1.93$ | 73.39 |
| 0.5 , 0.05 , - | $70.83 \pm 1.35$ | $70.06 \pm 0.98$ | $89.25 \pm 0.38$ | $71.34 \pm 0.82$ | **75.37** |
| 0.5 , 0.01 , - | $71.05 \pm 1.62$ | $70.29 \pm 0.88$ | $89.44 \pm 0.36$ | $70.06 \pm 1.80$ | 75.21 |
| 0.5 , 0.001 , - | $71.72 \pm 0.77$ | $69.84 \pm 1.65$ | $88.88 \pm 0.42$ | $70.85 \pm 0.83$ | 75.32 |
| 0.5 , - , 0.5 | $68.92 \pm 0.59$ | $69.62 \pm 0.51$ | $89.99 \pm 0.38$ | $70.04 \pm 0.63$ | 74.74 |
| 0.5 , - , 0.1 | $71.04 \pm 0.96$ | $69.78 \pm 0.98$ | $89.68 \pm 0.51$ | $70.95 \pm 0.81$ | **75.36** |
| 0.5 , - , 0.05 | $71.59 \pm 1.01$ | $68.97 \pm 1.42$ | $89.57 \pm 0.23$ | $69.81 \pm 3.45$ | 74.99 |
| 0.5 , 0.05 , 0.1 | $71.09 \pm 1.10$ | $69.55 \pm 0.54$ | $89.56 \pm 0.33$ | $71.31 \pm 0.90$ | 75.37 |
| 0.5 , 0.01 , 0.1 | $70.91 \pm 0.81$ | $70.05 \pm 1.33$ | $89.80 \pm 0.44$ | $71.46 \pm 0.46$ | 75.56 |
| 0.5 , 0.005 , 0.1 | $70.95 \pm 0.77$ | $69.78 \pm 0.91$ | $89.56 \pm 0.64$ | $71.00 \pm 1.12$ | 75.32 |
| 0.5 , 0.05 , 0.05 | $70.55 \pm 1.17$ | $69.57 \pm 1.14$ | $89.33 \pm 0.55$ | $70.40 \pm 2.88$ | 74.96 |
| 0.5 , 0.01 , 0.05 | $71.34 \pm 0.87$ | $70.29 \pm 0.77$ | $89.92 \pm 0.42$ | $71.15 \pm 1.02$ | **75.67** |
| 0.5 , 0.005 , 0.05 | $70.51 \pm 2.26$ | $69.60 \pm 0.58$ | $89.69 \pm 0.39$ | $71.51 \pm 0.84$ | 75.33 |

Table 6: Results of deeper networks on PACS dataset with object recognition accuracy (%) averaged over 5 runs.

| Method | Art Painting | Cartoon | Photo | Sketch | Average |
|---|---|---|---|---|---|
| | | ResNet-18 | | | |
| DeepAll | $78.93 \pm 0.46$ | $75.02 \pm 0.89$ | $96.60 \pm 0.16$ | $70.48 \pm 0.84$ | 80.25 |
| Basic-Adv | $80.54 \pm 1.71$ | $75.21 \pm 0.92$ | $\mathbf{96.67 \pm 0.21}$ | $70.65 \pm 1.91$ | 80.77 |
| Ours | $\mathbf{80.70 \pm 0.71}$ | $\mathbf{76.40 \pm 0.34}$ | $96.65 \pm 0.21$ | $\mathbf{71.77 \pm 1.27}$ | **81.38** |
| | | ResNet-50 | | | |
| DeepAll | $86.18 \pm 0.34$ | $76.79 \pm 0.33$ | $98.14 \pm 0.15$ | $74.66 \pm 0.93$ | 83.94 |
| Basic-Adv | $87.11 \pm 1.08$ | $78.65 \pm 1.13$ | $98.22 \pm 0.17$ | $\mathbf{76.48 \pm 1.09}$ | 85.11 |
| Ours | $\mathbf{87.51 \pm 1.03}$ | $\mathbf{79.31 \pm 1.40}$ | $\mathbf{98.25 \pm 0.12}$ | $76.30 \pm 0.65$ | **85.34** |

all scene-centric. This makes the generalization of the model difficult, although the domain shifts in VLCS are small [28]. In contrast, the images in all domains of PACS are mostly object-centric, and objects in different domains mainly have different styles and shapes. This can better evaluate the generalization capabilities of the model.

## 3.3 Ablation Studies

The experimental results above have demonstrated the effectiveness of our proposed algorithm for domain generalization. Here, we provide the ablation studies on the designed loss and network backbone to analyze the contributions of the proposed entropy regularization further.

**Different Weighting Factors.** We conduct various experiments with different weighting factors on PACS to examine their impacts. We report the average accuracy of 5 trials in Table 5. The results marked by the "gray" color correspond to the results reported in Table 4. "-" means the corresponding loss term is ignored. As shown in Table 5, in most cases, our proposed conditional entropy regularization ($\alpha_2 \neq 0$) can yield some improvements. Besides, by optimizing the full objective, our approach can further improve the generalization capabilities of the model.

**Deeper Networks.** We further study the generalization capabilities of our model by taking deeper networks, *e.g.,* ResNet-18 and RestNet-50 [41], as the backbone network. The models are pre-trained on ImageNet, and fine-tuned on PACS using the proposed loss. In specific, we take the last FC layer as our task network $T$, and other layers as the feature extractor $F$. We use three FC layers with output dimensions of 1024, 256, and the number of source domains / categories to construct the discriminator $D$ and classifiers $\{T_i, T_i'\}_{i=1}^3$, respectively. For both ResNet-18 and ResNet-50, we use the same hyper-parameters, *i.e.,* $\alpha_1 = 0.1$, $\alpha_2 = 0.001$, $\alpha_3 = 0.05$, and the learning rate of $1e-3$ ($F, T, D$) and $1e-4$ ($\{T_i, T_i'\}_{i=1}^3$). We learn models for 100 epochs, and report the average scores of 5 trials. As shown in Table 6, even though we take deeper networks as our backbones, our approach still yield higher scores than the two baselines.

**Class Imbalance.** We address the class imbalance issue by using the weighted cross-entropy loss according to the number of each class in each batch. If not using the weighted loss *i.e.,* setting the

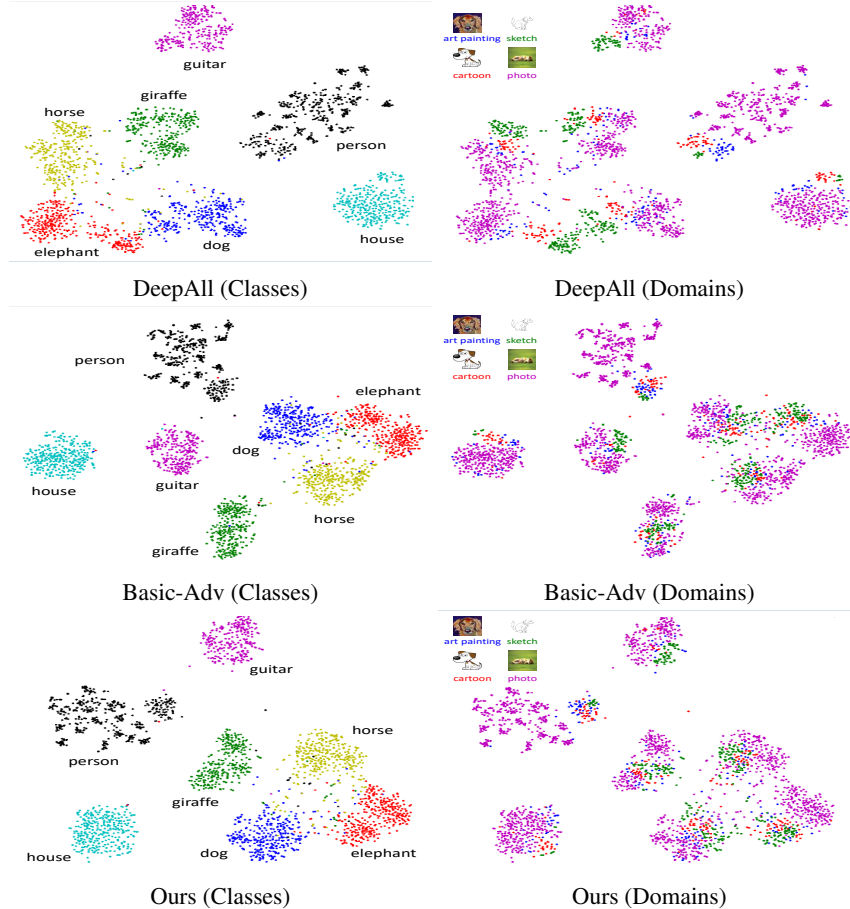

Figure 2: Feature visualization. Left: different colors represent different classes; Right: different colors indicate different domains (Target: Photo). Best viewed in color (Zoom in for details).

weight to 1 for each class, the model yields a lower average accuracy of $75.58\%$ (weighted loss used: $75.67\%$) on PACS, but still has better generalization capabilities.

**Feature Visualization.** To better understand the distribution of the learned features, we exploit t-SNE [42] to analyze the feature space learned by DeepAll, Basic-Adv, and Ours, respectively. We conduct this study on PACS, and in specific, we take the Photo dataset as the target, and others as the source. As shown in Figure 2, both Ours and Basic-Adv are capable of minimizing the distance between the distributions of the domains. For example, in DeepAll (Domains), we can observe that the Sketch (Green) is far away from other domains, while in Ours (Domains) and Basic-Adv (Domain), domains are clustered better. Furthermore, the comparison between Ours (Classes, Domains) and Basic-Adv (Classes, Domains) can show that our approach also discriminates the data from different categories better than Basic-Adv.

## 4  Conclusion

In this paper, we aim at learning the domain-invariant conditional distribution, which the basic adversarial learning based solutions cannot reach. We analyze the issues existed in related works, and propose an entropy regularization term, *i.e.,* the conditional entropy $H(Y|F(X))$, as the remedy. Our approach can produce domain-invariant features by optimizing the proposed regularization term coupled with the cross-entropy loss and the domain adversarial loss, and thus has a better generalization capability. The experimental results on both simulated and real-world datasets demonstrate the effectiveness of our proposed method. In the future, we can extend our approach to other challenging tasks, like semantic segmentation.

# 5 Acknowledgement

This research was supported by Australian Research Council Projects FL-170100117, DP-180103424, IH-180100002, and DE190101473.

## Broader Impact

Model generalization is a significant subject, since it is almost impossible for us to train a model for each scenario. However, due to the domain bias, the model trained on a domain often performs worse on other domains. Through exploiting the domain generalization techniques, we can train a model on the publicly available datasets, and then deploy it on other related scenarios directly or with few adaptations. Therefore, the industries can reduce their costs in repeating training the models. On the other hand, since the model is trained on multiple datasets sampled from different domains, the domain generalization techniques can reduce over-fitting, and thus courage the model generate fair results. Based on our knowledge, our work may not have an adverse impact on ethical aspects and future societal consequences.

## Footnotes

[1]Source/Target: seen/unseen during training.

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
