[Supplementary Material]

# Domain Generalization via Entropy Regularization
## -Supplementary Materials-

**Shanshan Zhao**
The University of Sydney
Australia
szha4333@uni.sydney.edu.au

**Mingming Gong**
University of Melbourne
Australia
mingming.gong@unimelb.edu.au

**Tongliang Liu**
The University of Sydney
Australia
tongliang.liu@sydney.edu.au

**Huan Fu**
Alibaba Group
China
fuhuan.fh@alibaba-inc.com

**Dacheng Tao**
The University of Sydney
Australia
dacheng.tao@sydney.edu.au

Here, we provide the proofs and the illustration of our framework supporting the contents in the submission.

## S. 1 Proof of Theorem 1

*Proof.* According to the definition of mutual information and under the assumption that all classes are equally likely, we have:

$$
\begin{aligned}
& - H_{P_i}(Y|F(X)) \\
=& I_{P_i}(Y, F(X)) - H(Y) \\
=& H_{P_i}(F(X)) - H_{P_i}(F(X)|Y) - H(Y) \\
=& - \frac{1}{C} \sum_{c=1}^{C} \mathop{\mathbb{E}}_{X' \sim P_i^F(X|Y)} \log P_i(X') + \frac{1}{C} \sum_{c=1}^{C} \mathop{\mathbb{E}}_{X' \sim P_i^F(X|Y)} \log P_i(X'|Y=c) - H(Y) \\
=& \frac{1}{C} \sum_{c=1}^{C} \mathop{\mathbb{E}}_{X' \sim P_i^F(X|Y)} \log \frac{P_i(X'|Y=c)}{P_i(X')} - H(Y) \\
=& \frac{1}{C} \sum_{c=1}^{C} KL(P_i(X'|Y=c)||P_i(X')) - H(Y) \\
=& \frac{1}{C} \sum_{c=1}^{C} KL(P_i(F(X)|Y=c)||P_i(F(X))) - H(Y) \\
=& JSD(P_i(F(X)|Y=1), P_i(F(X)|Y=2), \cdots, P_i(F(X)|Y=C)) - H(Y).
\end{aligned}
\tag{S. 1}
$$

Since $H(Y)$ is a constant, then minimizing $-H_{P_i}(Y|F(X))$ is equivalent to minimizing $JSD(P_i(F(X)|Y=1), P_i(F(X)|Y=2), \cdots, P_i(F(X)|Y=C))$, the global minimum of which is achieved at $P_i(F(X)|Y=1) = P_i(F(X)|Y=2) = \cdots = P_i(F(X)|Y=C)$. $\qquad\square$

## S. 2 Proof of Theorem 2

**S. Proposition 1.** *Let* $V(F, \{T'_i\}) = \sum_{i=1}^{K} \mathbb{E}_{(X,Y) \sim P_i(X,Y)} [\log Q_i^{T'_i}(Y|F(X))]$. *Then the optimal prediction probabilities of* $T'_i$ *are*

$$\langle T'^*_i(\mathbf{x}'_i) \rangle_c = Q_i^{T'^*_i}(Y = c|\mathbf{x}'_i) = \frac{P_i(\mathbf{x}'_i|Y = c)}{\sum_{c=1}^{C} P_i(\mathbf{x}'_i|Y = c)}, \tag{S. 2}$$

*where* $\langle \mathbf{z} \rangle_i$ *denotes the* $i^{th}$ *element of* $\mathbf{z}$*, and* $\mathbf{x}'_i = F(\mathbf{x}_i)$*.*

*Proof.* For a fixed $F$, $\min_F \max_{\{T'_i\}} V(F, \{T'_i\})$ reduces to maximizing $V(F, \{T'_i\}_{i=1}^{K})$ w.r.t. $\{T'_1, T'_2, \cdots, T'_K\}$[1]:

$$\{\langle T'^*_i(\mathbf{x}') \rangle_1, \langle T'^*_i(\mathbf{x}') \rangle_2, \cdots, \langle T'^*_i(\mathbf{x}') \rangle_C\}$$

$$= arg \max_{\{\langle T'_i(\mathbf{x}) \rangle_c\}_{c=1}^{C}} \sum_{c=1}^{C} \int_{\mathbf{x}'_i} P_i(\mathbf{x}'_i|Y = c) \log(\langle T'_i(\mathbf{x}'_i) \rangle_c) d\mathbf{x}'_i, \tag{S. 3}$$

$$s.t. \sum_{c=1}^{C} \langle T'_i(\mathbf{x}'_i) \rangle_c = 1.$$

Maximizing the value function point-wisely and applying Lagrange multipliers, we obtain the following problem:

$$\{\langle T'^*_i(\mathbf{x}') \rangle_1, \langle T'^*_i(\mathbf{x}') \rangle_2, \cdots, \langle T'^*_i(\mathbf{x}') \rangle_C\}$$

$$= arg \max_{\{\langle T'_i(\mathbf{x}') \rangle_c\}_{c=1}^{C}} \sum_{c=1}^{C} P_i(\mathbf{x}'_i|Y = c) \log(\langle T'_i(\mathbf{x}'_i) \rangle_c) + \lambda_i(\sum_{c=1}^{C} \langle T'_i(\mathbf{x}'_i) \rangle_c - 1). \tag{S. 4}$$

Setting the derivative of Eq. S. 4 w.r.t. $\langle T'_i(\mathbf{x}'_i) \rangle_c$ to zero, we obtain $\langle T'^*_i(\mathbf{x}_i) \rangle_c = -\frac{P_i(\mathbf{x}'_i|Y=c)}{\lambda_i}$. Through substituting the value of $\langle T'^*_i(\mathbf{x}_i) \rangle_c$ into the constraint $\sum_{c=1}^{C} \langle T'_i(\mathbf{x}'_i) \rangle_c = 1$, we can obtain $\lambda_i = -\sum_{c=1}^{C} P_i(\mathbf{x}'_i|Y = c)$, and thus get the optimal solution $\langle T'^*_i(\mathbf{x}'_i) \rangle_c = \frac{P_i(\mathbf{x}'_i|Y=c)}{\sum_{c=1}^{C} P_i(\mathbf{x}'_i|Y=c)}$. □

**S. Theorem 1.** *If* $U(F)$ *is the maximum value of* $V(F, \{T'_i\}_{i=1}^{K})$*, i.e.,*

$$U(F) = \sum_{i=1}^{K} \sum_{c=1}^{C} \mathbb{E}_{X_i \sim P_i(X)} [\log \frac{P_i(X'_i|Y = c)}{\sum_{c=1}^{C} P_i(X'_i|Y = c)}], \tag{S. 5}$$

*the global minimum of the minimax game is attained if and only if* $P_i(X'_i|Y = 1) = P_i(X'_i|Y = 2) = \cdots = P_i(X'_i|Y = C)$ *for any* $i \in \{1, 2, \cdots, K\}$*, where* $U(F)$ *achieves the value* $-KC \log C$*.*

*Proof.* Adding $KC \log C$ to $U(F)$ can obtain:

$$U(F) + KC \log C = \sum_{i=1}^{K} \sum_{c=1}^{C} \{ \mathbb{E}_{X_i \sim P_i(X)} [\log \frac{P_i(X'_i|Y = c)}{\sum_{c=1}^{C} P_i(X'_i|Y = c)}] + \log C \}$$

$$= \sum_{i=1}^{K} \sum_{c=1}^{C} \mathbb{E}_{X_i \sim P_i(X)} [\log \frac{P_i(X'_i|Y = c)}{\frac{1}{C} \sum_{c=1}^{C} P_i(X'_i|Y = c)}] \tag{S. 6}$$

$$= \sum_{i=1}^{K} \sum_{c=1}^{C} KL(P_i(X'_i|Y = c)||\frac{1}{C} \sum_{c=1}^{C} P_i(X'_i|Y = c)).$$

According to the definition of the Jensen-Shannon divergence, we can obtain $U(F) = -KC \log C + \sum_{i=1}^{K} C \cdot JSD(P_i(X'_i|Y = 1), P_i(X'_i|Y = 2), \cdots, P_i(X'_i|Y = C))$. Since the JSD between

S. Figure 1: Illustration of our framework. GRL represents the gradient reversal layer. All components are trained, but only $F$ and $T$ are preserved for test.

multiple distributions is always non-negative, and zero iff they are equal, then we have

$$P_1(X_1'|Y=1) = P_1(X_1'|Y=2) = \cdots = P_1(X_1'|Y=C),$$
$$P_2(X_2'|Y=1) = P_2(X_2'|Y=2) = \cdots = P_2(X_2'|Y=C),$$
$$\cdots$$
$$P_K(X_K'|Y=1) = P_K(X_K'|Y=2) = \cdots = P_K(X_K'|Y=C),$$

(S. 7)

and the global minimum of $U(F)$ is $-KC \log C$. $\qquad\square$

## S. 3 Framework

Here, we provide an illustration of our framework in S. Figure 1 for better understanding of the proposed components. The main module consists of a feature extractor $F$ and a classifier $T$. In addition, we exploit a domain discriminator $D$ to discriminate domains, and $2K$ classifiers ($\{T_i\}_{i=1}^K$ and $\{T_i'\}_{i=1}^K$) to regularize the generated features. We insert a gradient reversal layer (GRL) [1] between $F$ and $D$, and $F$ and $T_i'$, respectively. In the inference stage, only the main module ($F$ and $T$) is required.

## Footnotes

[1]Here, we only consider $T'_i$ for simplicity.

## References

[1] Yaroslav Ganin and Victor S. Lempitsky. Unsupervised domain adaptation by backpropagation. In *ICML*, 2015.