[Reviews · NeurIPS 2020]

Review 1

Summary and Contributions: Authors address an essential problem in domain generalization - to learn discriminative domain-invariant features. Authors use a combination of entropy regularization loss along with usual loss like cross entropy loss for classification and adversarial loss for domain discrimination. Authors compare their method on various domain generalization datasets and state-of-the-art methods.

Strengths: 1) Authors have pointed out the problem of learning discriminative domain-invariant features clearly. Simulated data shown in figure 1 clarifies the problem that authors are trying to solve. 2) Authors show results on 2 simulated datasets and 2 real world datasets. All these datasets are used to benchmark domain generalization methods. 3) Each loss in Equation 10 is explained and intuition given on why each of the terms is important.

Weaknesses: 1) Can authors give Architecture size/number of parameters to learn for state-of-the-art methods and proposed method? 2) How were hyperparameters selected? Was there a train, validation and test set? 3) Some of the state-of-the-art methods (especially ones with sound theory) were completely ignored. Can authors add these papers at least in discussion if not in experiments [1-3]? 4) Dataset presented in Figure 1 is a really good motivation. Could authors get results on such a dataset for various state-of-the-art methods and proposed methods? [1] Blanchard, Gilles, et al. "Domain generalization by marginal transfer learning." arXiv preprint arXiv:1711.07910 (2017). This is an extended paper from previous NIPS 2011 paper. [2] Muandet, Krikamol, David Balduzzi, and Bernhard Schölkopf. "Domain generalization via invariant feature representation." International Conference on Machine Learning. 2013. [3] Deshmukh, Aniket Anand, et al. "A Generalization Error Bound for Multi-class Domain Generalization." arXiv preprint arXiv:1905.10392 (2019). Reviewers have addressed my concerns and clarified most of the doubts I had. I am changing my score from 6 to 7.

Correctness: Looks correct.

Clarity: Yes.

Relation to Prior Work: Some of the theoretically sound paper were omitted from the discussion completely.

Reproducibility: Yes

Additional Feedback:


Review 2

Summary and Contributions: Domain generalization is a challenging problem. This paper proposes a regularization approach using entropy loss and a few other conventional loss functions for generalization across domain shift. Thus in addition to cross entropy loss for classification, and adversarial loss for domain discrimination, the overall objective is to learn conditional-invariant features across all source domains. Thus the proposed method learns classifiers with better generalization capabilities. Experimental results on many datasets are provided.

Strengths: The formulation is theoretically sound and supported by well-reasoned theorems. The authors have compared with several existing methods on multiple datasets. Ablation studies are also strong.

Weaknesses: I am old school. So I am not happy when small percentage improvements are marketed as outperforming, superior etc. If you really look at the tables, the results are somewhat mixed. The proposed method is not uniformly better than existing methods. For example, in Table 3 only for Pascal VOC the proposed methods is performing better. For other datasets, the improvements are either small or non-existent. Likewise, in Table 4, the proposed method does not improve on SOTA for cartoon and photo domains.

Correctness: No and Yes. Claims of outperforming and superiority are not fully justified.

Clarity: Yes.

Relation to Prior Work: Yes

Reproducibility: Yes

Additional Feedback: This is a good paper, although performance improvement is not uniformly better than SOTA. I will suggest to the authors that they tone down the claims! I read the other reviews and the author's rebuttal. I am satisfied with the rebuttal and will keep my current recommendation.


Review 3

Summary and Contributions: Authors propose to tackle the problem of domain generalization by finding domain-invariant feature representations across domains with invariant conditional distributions. It's an extension of the adversarial learning approach to domain generalization. Authors accomplish this by minimizing the Jensen-Shannon divergence among conditional distributions under the pressumption that it is equivalent to entropy regularization. Authors show that this approach improves domain generalization among several datasets.

Strengths: - The paper is well written and easy to follow. - The approach is theoretically sound - Authors perform exhaustive evaluation across multiple datasets - Results presented show strong benefits for some of the datasets tested performance wise of using entropy regularization over basic adversarial learning - Authors provided code to reproduce experiments

Weaknesses: - The entire approach relies under the assumption the minimizing JSD is equivalent to the intended entropy regularization. This is only true when classes are perfectly balanced. It would be interesting to see how much would the performance be affected based on class unbalance. - The approach requires an aditional classifier per domain/dataset which is a bit worrisome to me. How much of the improvement is coming from extra capacity given by extra classifiers? How does the model perform if you remove then? How well you do if the last term of the loss in equation 10 is removed? What happens when the number of domains is large? - For many of the datasets tested the improvement over other approaches or even the general adversarial approach is marginal Post Rebuttal: Authors clarified most of my concerns so I am raising the score.

Correctness: Claims, approach, and empirical methodolody are correct.

Clarity: The paper is very well written and easy to follow. I found minor grammatical mistakes on lines 31, 60/61.

Relation to Prior Work: Authors properly addressed related work and how their approach is different to others previously proposed.

Reproducibility: Yes

Additional Feedback: I like the way authors presented their work and how they tested the approach in several benchmarks. I think that showing how inbalance affect the approach and to what extend patch re-weighting alleviates the issue is important. Show how much of the gain in performance is coming from minimizing KL vs having additional classifiers (more capacity) is also important.

[Author Response · NeurIPS 2020]

We sincerely thank all reviewers for their valuable comments. Below we address concerns raised by all reviewers. We
will carefully revise our paper, and release the code provided in the submission for reproducibility.

**\*Q1 (R1) The number of parameters. A1:** As stated in Section 3 in the submission, we follow the existing works and
evaluate our method using AlexNet and ResNet. Therefore, our model has **the same number of parameters as the**
**other methods**, *i.e.,* ~61M (AlexNet) and ~11M (ResNet-18) during inference. Moreover, at the training stage, most
methods exploit auxiliary modules to model the constraints, which increase the number of parameters. For example,
MASF [6] proposes a metric-learning component with two fully-connected layers for local sample clustering. Apart
from the main network, Epi-FCR [20] trains one additional feature extractor and classifier for each domain. We provide
the performance and the number of parameters (AlexNet) of several methods in Table 1 for better comparison.

Table 1: Performance on PACS, additional parameters at training, and parameters at inference. Left to right: Epi-FCR [20], CIDDG [22], MASF [6], and Ours.

| Accuracy/% | | | | Additional Param/M (Train) | | | | Param/M (Infer) | | | |
|---|---|---|---|---|---|---|---|---|---|---|---|
| 72.03 | 72.20 | 75.21 | 75.67 | 183 | 83.89 | 4.46 | 73.4 | 61 | 61 | 61 | 61 |

**\*Q2 (R1) Hyperparameters and dataset splits. A2:** At the training stage, we train the model on **the training set of**
**the source domains**, and choose the hyperparameters and the best model on **the validation set** of the source domains.
To evaluate the generalization capabilities, we **test the best model selected by source domain validation data** on the
target dataset (whole set for VLCS, official test set for PACS). We will clarify this in the final version.

**\*Q3 (R1) Missing some existing works. A3:** Deshmukh *et al.* [3] proposed the kernel-based method for multi-class
domain generalization, and proved the first known generalization error bound . Blanchard *et al.* [1] analyzed the problem
of domain generalization from a different perspective by augmenting the original feature space. Then, they developed a
kernel-based method that predicts classifiers from augmented feature space. As stated in Section 1 of the submission,
Muandet *et al.* [2] proposed a kernel-based optimization algorithm, called DICA, which can not only minimize the
difference between marginal distributions of the domains but also preserve the functional relationship between input and
output variables. **In contrast, our work focuses on learning conditional-invariant deep representations across all**
**source domains. We will add the discussion in the revision.**

**\*Q4 (R1) Dataset in Figure 1. A4:** Here, we compare our method with two methods, *i.e.,* the basic solution through
adversarial learning (Basic-Adv) and CIDDG [22]. The latter one aims to learn domain-invariant features by introducing
one domain discriminator for each class. To create the target dataset, we slightly adjust the two marginal distributions
of Domain_0. The average accuracy over 5 repeated experiments is 78.2% (Ours), 78.0% (CIDDG), and 77.6%
(Basic-Adv), respectively. Ours and CIDDG have close performance on this simple simulation dataset, since they both
focus on learning the domain-invariant features. Additionally, both of them perform better than the baseline adversarial
training method.

**\*Q5 (R3&R4) Claims about the improvements. A5:** In comparison to existing methods and the provided strong
baseline, our model overall performs better. For example, our method yields higher average accuracy on both VLCS
and PACS. For the Cartoon and Photo datasets, we hypothesize that the distinctive shape on the former and background
on the latter make our method less effective on the two datasets. We appreciate the reviewer's constructive comments
on the claim, and we will carefully improve the presentation in the revision.

**\*Q6 (R4) Class imbalance. A6:** We address the class imbalance issue by using the weighted cross-entropy loss accord-
ing to the number of each class in each batch, which can be found in the provided source code (func *_compute_cls_loss*
in train.py). **If not using the weighted loss**, *i.e.,* setting the weight to 1 for each class, the model yields a lower average
accuracy of 75.58% (weighted loss used: 75.67%) on PACS.

**\*Q7 (R4) Additional classifiers. A7:** We exploit **the additional classifiers only at the training stage and remove**
**them during inference,** *i.e.,* **only the feature extractor $F$ and the main classifier $T$ are preserved**, which can be
found in the submitted supplementary (S.Section 3 and S.Figure 1). Therefore, our model has the same capacity as
other methods in the inference stage, please refer to **\*Q1** and Table 1 for details of the model parameters . Moreover,
we have analyzed the loss function in the ablation study (Section 3.3). Specifically, when removing the last term of the
loss function (*i.e.,* removing the extra classifiers), we obtain an average accuracy of 75.37% ($\alpha_3 = 0$ in Table 5), **which**
**is lower than using all terms but higher than no entropy regularization** ($\alpha_2 = 0$). As we stated in L133-136 of the
submission, we use the extra classifiers to make the training stage more stable. Additionally, as shown in Table 5, the
model trained with the extra classifiers but without the proposed entropy regularization does not perform well in most
cases (*e.g.,* the accuracy is less than 75%), while the entropy regularization performs better (>75.2%). This also shows
the significance of the entropy regularization. The number of domains would affect the consumed memory during
training, but has no impact in inference. **In a nutshell, the extra classifiers are only adopted at the training stage**
**for improving the stability, and do not increase the number of parameters and model complexity. Additionally,**
**the effectiveness of the proposed regularization term is verified in the ablation study.**

[Meta-Review · NeurIPS 2020]

This paper presents an interesting perspective to the domain generalization problem, proposing to learn representations that are condtional-invariant across the source datsets. Analysis and theory on simple datasets are used to motivate the problem and approach, and results are shown on larger datasets. Some reviewer concerns were expressed in terms of the methodology and claims, but the rebuttal largely addressed these. Overall, the paper presents a well-reasoned approach that is theoretically-motivated. The authors should make sure to especially calibrate their empirical claims based on the reviewer feedback.